# Tabular Transfer Learning via Prompting LLMs

**Jaehyun Nam[1], Woomin Song[1], Seong Hyeon Park[1], Jihoon Tack[1]**
**Sukmin Yun[2], Jaehyung Kim[3], Kyu Hwan Oh[4], Jinwoo Shin[1]**
[1]KAIST, [2]Hanyang Unviersity ERICA, [3]Carnegie Mellon University
[4]Hyundai Motor Company
{jaehyun.nam, jinwoos}@kaist.ac.kr

## Abstract

Learning with a limited number of labeled data is a central problem in real-world applications of machine learning, as it is often expensive to obtain annotations. To deal with the scarcity of labeled data, transfer learning is a conventional approach; it suggests to learn a transferable knowledge by training a neural network from multiple other sources. In this paper, we investigate transfer learning of tabular tasks, which has been less studied and successful in the literature, compared to other domains, *e.g.*, vision and language. This is because tables are inherently heterogeneous, *i.e.*, they contain different columns and feature spaces, making transfer learning difficult. On the other hand, recent advances in natural language processing suggest that the label scarcity issue can be mitigated by utilizing in-context learning capability of large language models (LLMs). Inspired by this and the fact that LLMs can also process tables within a unified language space, we ask whether LLMs can be effective for tabular transfer learning, in particular, under the scenarios where the source and target datasets are of different format. As a positive answer, we propose a novel tabular transfer learning framework, coined *Prompt to Transfer (P2T)*, that utilizes unlabeled (or heterogeneous) source data with LLMs. Specifically, P2T identifies a column feature in a source dataset that is strongly correlated with a target task feature to create examples relevant to the target task, thus creating pseudo-demonstrations for prompts. Experimental results demonstrate that P2T outperforms previous methods on various tabular learning benchmarks, showing good promise for the important, yet underexplored tabular transfer learning problem. Code is available at https://github.com/jaehyun513/P2T.

## 1 Introduction

Learning with a limited number of labeled samples is often a critical requirement in real-world machine learning applications. This limited data problem is particularly important in the tabular domain; tabular datasets often require substantial annotation efforts (*e.g.*, credit risk assessment; Clements et al., 2020), or it is hard to obtain new samples for emerging tasks (*e.g.*, identifying patients with new diseases such as COVID-19; Peplow, 2016; Zhou et al., 2020). To address this problem, various methods have been thoroughly studied in domains such as vision (Assran et al., 2021; Pham et al., 2021) and language (Chen et al., 2021; Min et al., 2022). However, the research on tabular data has only recently begun to gain traction (Yoon et al., 2020; Nam et al., 2023), despite its wide-ranging impact across a variety of industries (Guo et al., 2017; Ulmer et al., 2020; Zhang et al., 2020).

Learning transferable knowledge by training a neural network from various sources (Chen et al., 2020; Perez et al., 2021; Lee & Shin, 2022) is a common way to address this limited data problem in other domains. However, such transfer learning is challenging in the tabular domain, because the source and target datasets are often very heterogeneous (*i.e.*, they have different columns and feature spaces; Zhu et al., 2023; Wang & Sun, 2022; Yan et al., 2024). For example, using the collected features to predict diabetes to predict whether another

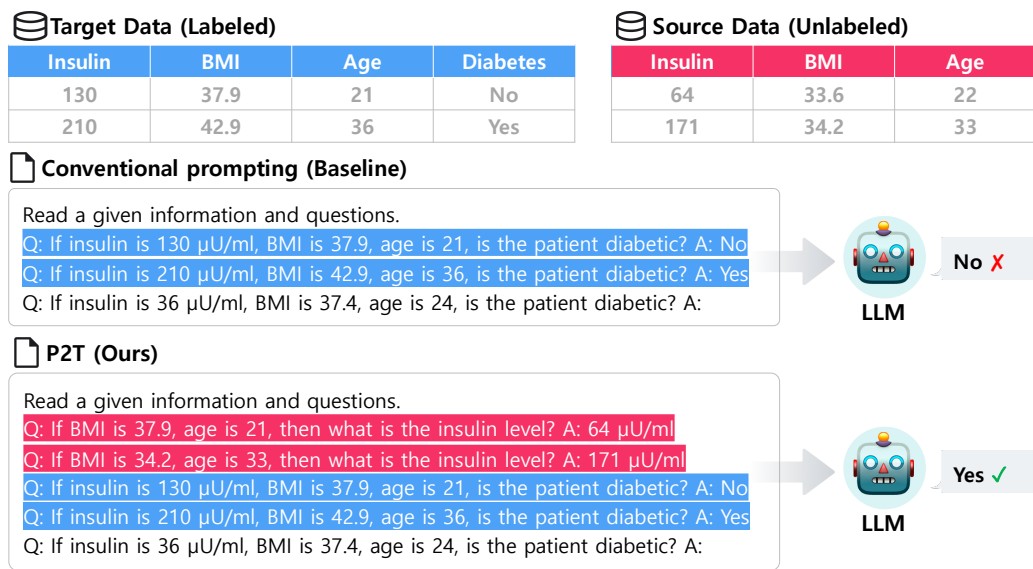

Figure 1: **Overview.** P2T creates pseudo-demonstrations to effectively transfer knowledge of source data in an in-context manner. 'Insulin' is used as a prediction target for pseudo-demonstrations because it has the highest correlation with the task feature 'Diabetes.'

patient has breast cancer is not straightforward because the required features to predict each disease (*i.e.*, columns in a table) are very different. This leads to the question of *how transferable knowledge can be extracted from various tabular data sources in a unified space.*

On the other hand, converting tabular data into text has recently gained attention as this conversion leads to several benefits; first, one can inject linguistic context (Dinh et al., 2022; Hegselmann et al., 2023; Manikandan et al., 2023), for example, using column name 'Age' along with its numerical value. In addition, it opens up using large language models (LLMs) for tabular learning, which have demonstrated the capability to address limited data problems with few task-specific instructions (Brown et al., 2020; Dong et al., 2022; Wei et al., 2022; Kim et al., 2024), known as *prompting* (or *in-context learning; ICL*).

**Contribution.** Motivated by this, we ask whether LLMs can be tabular transfer modules and address limited data problems in the tabular domain. As a positive answer, we propose a novel tabular transfer learning framework that utilizes LLM's ICL capabilities, coined *Prompt to Transfer (P2T)*; see the overview in Figure 1. Our main idea is to use LLM to extract transferable knowledge from the source dataset (*e.g.*, unlabeled dataset) and use it as in-context samples when prompting the LLM (called *pseudo-demonstrations*).

Specifically, P2T starts by prompting LLM to determine which column feature is most important for the target task based on given few-shot labeled samples (*i.e.*, demonstrations). P2T then creates pseudo-demonstrations that describe the prediction task where the selected column feature is the target and the remaining ones are input, thus ensuring high relevance to the actual target task. These pseudo-demonstrations reveal how to predict the selected column feature from the remaining features and this knowledge would be useful to the original target task (*e.g.*, knowledge to predict 'Insulin' could be useful to predict 'Diabetes'). Therefore, P2T prompts the LLM to generate the desired output through the created pseudo-demonstrations with few-shot labeled demonstrations.

We verify the effectiveness of P2T, by conducting comprehensive evaluations on diverse tabular learning scenarios considering different source types (*i.e.*, unlabeled and heterogeneous datasets) and the number of labeled samples (*i.e.*, few-shot and zero-shot). Our results show that P2T significantly and consistently outperforms existing methods, including self-supervised (Yoon et al., 2020) and unsupervised meta-learning (Nam et al., 2023) methods, by transferring knowledge through prompting. As LLMs recently continue to advance (*e.g.*, GPT-4; OpenAI, 2023), improved performance is expected with future models.

## 2 Related work

**Tabular transfer learning.** Researchers have developed a number of methods to train transferable representations for tabular data. One major stream of work exploits unlabeled datasets. In this context, Yoon et al. (2020) and Ucar et al. (2021) introduced various pre-text task losses (*e.g.*, reconstruction loss) with the goal of self-supervised learning on tabular datasets. In addition, Nam et al. (2023) proposed an unsupervised meta-learning framework to address few-shot tabular learning problems. Another stream of work exploits multiple heterogeneous datasets. Zhu et al. (2023) used a federated learning approach with separate encoders for each dataset to aggregate the parameters into a single transformer. Wang & Sun (2022) and Yan et al. (2024) used tokenizers to map the heterogeneous structure of tables into a unified language space and then fine-tuned the language model. We propose a transfer learning framework that leverages both unlabeled and heterogeneous sources while applying in-context transfer to enable immediate predictions (*i.e.*, training-free).

**Tabular learning with large language models.** Recent advances in LLMs have provided an impetus to explore their potential for tabular learning. Dinh et al. (2022) investigated the performance of fine-tuned GPT-3 models (Brown et al., 2020) on tabular data. Extending this line of research, Hegselmann et al. (2023) conducted a comprehensive analysis using the T0 model (Sanh et al., 2022), leveraging the language prior in LLMs. Their analysis is extended to sample efficiency, even conducting zero-shot experiments. Recently, Manikandan et al. (2023) developed a boosting framework over prompting LLMs that uses LLMs as weak learners in tabular prediction tasks. Inspired by these preceding studies, our work proposes a method for the effective exploitation of various transfer sources - an aspect overlooked in prior research. By integrating the utilization of unlabeled (or heterogeneous) data with LLMs, we aim to enhance performance in transfer learning scenarios.

**In-context learning.** As model and dataset sizes increase (Brown et al., 2020), LLMs have exhibited the capability for ICL (or prompting), where they draw knowledge from a handful of contextual examples. For example, Wei et al. (2022) have illustrated the competency of LLMs in solving mathematical reasoning problems via ICL. The ICL process begins by employing a small number of examples to establish a contextual framework, typically constructed using natural language templates. Following this, a query question and a contextual demonstration are combined to form a prompt, which is subsequently fed to the LLMs for prediction. Notably, ICL does not necessitate parameter updates and directly carries out predictions using LLMs, enabling easy implementation for real-world applications. In our work, we delve deeper into the potential of ICL by examining its performance on transfer learning, using source data for creating effective demonstrations.

## 3 P2T: Prompt to Transfer

In this section, we propose a simple yet effective tabular transfer learning that leverages ICL, the capability of LLMs that adapt to a new task using the context provided in a prompt without updating model parameters. In a nutshell, our framework creates pseudo-demonstrations from transfer sources that act as proxies for causal relationships between input features and labels, and then prompt the LLM to make predictions. We first briefly describe preliminaries (Section 3.1), and then the core component coined *Prompt to Transfer (P2T)*, which creates effective pseudo-demonstrations for ICL (Section 3.2).

### 3.1 Preliminaries

**Problem setup: tabular transfer learning.** We first describe the problem setup of our interest. A labeled target dataset $\mathcal{D}^t = \{(\mathbf{x}_i^t, \mathbf{y}_i^t)\}_{i=1}^{N^t} \subseteq \mathcal{X}^t \times \mathcal{Y}$ and a source dataset $\mathcal{D}^s = \{\mathbf{x}_i^s\}_{i=1}^{N^s} \subseteq \mathcal{X}^s$ are given, where $\mathbf{x}^t$ is $d^t$-dimensional, and $\mathbf{x}^s$ is $d^s$-dimensional feature which correspond to the value in the respective table columns. Also, we assume that column name sets $F = \{f_1, \cdots, f_d\}$ are given as well (*e.g.*, '$x_1$ : Male' is feature values of the '$f_1$ : sex' column), and $f_{target}$ is the column name of the labels $\mathbf{y}$ of the target dataset. Labels $\mathbf{y} \in \mathcal{Y}$ are provided in the form of natural language annotations (*e.g.*, 'Non Diabetic' and

'Diabetic' are labels of the Diabetes dataset). Here, the cardinality of $\mathcal{D}^t$ is assumed to be much smaller than the source dataset, *i.e.*, $|\mathcal{D}^t| \ll |\mathcal{D}^s|$. Our goal is to effectively extract knowledge from $\mathcal{D}^s$ to predict the accurate label of test query $\mathbf{x}_{test}^t$ in the target dataset.

**In-context learning for tabular prediction tasks.** We now describe ICL-based tabular prediction (Dinh et al., 2022). Similar to ICL in the natural language domain (Brown et al., 2020), tabular prediction via ICL is executed by constructing an input prompt comprising (i) a *task description* $p_{\texttt{task}}(\mathcal{D}^t)$, (ii) *few-shot labeled demonstrations* $p_{\texttt{label}}(\mathcal{D}^t)$ (*i.e.*, descriptions of input-output pairs of the labeled samples), and (iii) a *test query* $p_{\texttt{test}}(\mathbf{x}_{test}^t)$. To begin, the *task description* is manually crafted through a systematic procedure. This description includes prompts directing the LLM to analyze the data in a step-by-step manner, and the dataset's details such as column descriptions. Next, *few-shot labeled demonstrations* are created by serializing tabular data into natural language, thereby presenting the data in a question format. There exist numerous design choices for tabular serialization; however, we mainly follow Dinh et al. (2022). Finally, we predict the output by prompting LLM $\mathcal{M}$:

$$\mathbf{y}_{pred} = \mathcal{M}(p_{\texttt{task}}(\mathcal{D}^t) \oplus p_{\texttt{label}}(\mathcal{D}^t) \oplus p_{\texttt{test}}(\mathbf{x}_{test}^t)),$$

where, $\oplus$ is the concatenation of each prompt (see the *conventional prompting* in Figure 1).

## 3.2 In-context tabular transfer learning with P2T

We now present P2T, a novel approach to improve tabular prediction performance by creating additional *pseudo-demonstrations* from the source data, which serve as proxies for few-shot labeled demonstrations. This is achieved by (i) first identifying the column feature in the source dataset that exhibits the highest correlation with the target $f_{target}$ and (ii) then constructing *pseudo-demonstrations* from the transfer source. The constructed prompt $p_{\texttt{P2T}}$, which integrates *pseudo-demonstrations* with *conventional prompting*, is then fed into the LLM $\mathcal{M}$ to generate the desired prediction output.

**Correlation identification.** P2T begins by identifying the feature $f_k \in F^s$ (the column name set of the source data) that holds the most significant correlation with the task column $f_{target}$ in the target data $\mathcal{D}^t$. To achieve this, we ask LLM $\mathcal{M}$ which feature among $F^s$ is most important for predicting $f_{target}$. Formally, an input prompt designed to identify correlations consists of two main components: (i) *few-shot labeled demonstrations* $p_{\texttt{label}}(\mathcal{D}^t)$ and (ii) *identification instructions* $p_{\texttt{cor}}(F^s, f_{target})$; see Figure 2. Finally, we prompt LLM $\mathcal{M}$ to obtain $f_k$:

$$f_k = \mathcal{M}(p_{\texttt{label}}(\mathcal{D}^t) \oplus p_{\texttt{cor}}(F^s, f_{target})).$$

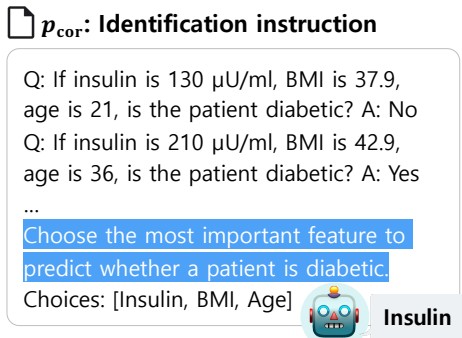

**📄 $p_{\text{cor}}$: Identification instruction**

Q: If insulin is 130 μU/ml, BMI is 37.9, age is 21, is the patient diabetic? A: No
Q: If insulin is 210 μU/ml, BMI is 42.9, age is 36, is the patient diabetic? A: Yes
...
Choose the most important feature to predict whether a patient is diabetic.
Choices: [Insulin, BMI, Age]        **Insulin**

Figure 2: Example prompt for correlation identification on the Diabetes dataset.

Our choice of using LLMs over conventional methods (Chen & Guestrin, 2016; Prokhorenkova et al., 2018) is driven by the capabilities of LLMs to interpret linguistic context; their ability to leverage both the semantics of column names and the associated numeric values offers a distinct advantage. For instance, considering a column like 'Age,' conventional algorithms would focus solely on the numeric values. Conversely, LLMs can utilize the semantic understanding of 'Age,' incorporating additional contextual information.

**Pseudo-demonstrations from the source tables.** Our main idea is to create *pseudo-demonstrations* from the unlabeled (or heterogeneous) source tables using highly correlated column feature $f_k$ as a new target feature. The rationale behind using $f_k$ stems from the intuition that predicting the most correlated column feature from the remaining features would resemble the original task of predicting the label from all the features; therefore, these pseudo-demonstrations from source data include the useful knowledge to predict the original target task. For instance, predicting 'Diabetes' from 'BMI' and 'Age' is similar to predicting 'Insulin' using the same features (Nam et al., 2023). Thus, we create *pseudo-demonstrations* that predict the value of $f_k$ from remaining features, which we refer to as

| Target dataset | Source dataset | Method | Accuracy (↑) |
|---|---|---|---|
| Adult | ✗ | zero-shot | 68.00 |
| | Credit-R | **P2T (Ours)** | 70.00 |
| | Electricity | **P2T (Ours)** | 72.00 |
| | Unlabeled Adult | **P2T (Ours)** | **74.00** |
| Credit-g | ✗ | zero-shot | 46.00 |
| | Credit-A | **P2T (Ours)** | 62.00 |
| | Unlabeled Credit-g | **P2T (Ours)** | **68.00** |
| Heart-c | ✗ | zero-shot | 60.00 |
| | Diabetes | **P2T (Ours)** | **65.00** |
| | Unlabeled Heart-c | **P2T (Ours)** | 63.33 |
| Breast | ✗ | zero-shot | 41.07 |
| | Haberman | **P2T (Ours)** | 58.93 |
| | Unlabeled Breast | **P2T (Ours)** | **62.50** |

Table 1: **Test accuracy (%) on various zero-shot learning scenarios.** Both unlabeled dataset and heterogeneous dataset improves the zero-shot test accuracy of the target dataset. Bold indicates the highest accuracy, and underlined indicates the second highest accuracy.

$p_{\mathtt{pseudo}}(\mathcal{D}^s, f_k)$. See Figure 1 for a specific example of how P2T creates *pseudo-demonstrations* from an unlabeled dataset. As shown in Figure 1, 'Inuslin' is selected as the most important feature $f_k$ for predicting whether a patient has diabetes (see Figure 2). Therefore, predicting insulin levels from BMI and age becomes components of *pseudo-demonstrations*. Finally, our P2T framework outputs prediction results as follows:

$$p_{\mathtt{P2T}}(\mathcal{D}^t, \mathcal{D}^s, f_k, \mathbf{x}^t_{test}) = p_{\mathtt{task}}(\mathcal{D}^t) \oplus p_{\mathtt{pseudo}}(\mathcal{D}^s, f_k) \oplus p_{\mathtt{label}}(\mathcal{D}^t) \oplus p_{\mathtt{test}}(\mathbf{x}^t_{test}),$$

$$\mathbf{y}_{pred} = \mathcal{M}(p_{\mathtt{P2T}}(\mathcal{D}^t, \mathcal{D}^s, f_k, \mathbf{x}^t_{test})).$$

## 4 Experiments

In this section, we validate the effectiveness of our proposed method for transfer learning scenarios on a variety of tabular datasets from the OpenML repositories (Vanschoren et al., 2014) and Kaggle. First, in Section 4.1, we verify that P2T improves zero-shot prediction performance by leveraging different types of transfer sources (*i.e.*, unlabeled and heterogeneous datasets). Note that zero-shot prediction is one of the big benefits of using LLMs that is not possible with traditional methods like CatBoost (Prokhorenkova et al., 2018). Then, in Section 4.2, we verify that our method outperforms other tabular learning methods, including unsupervised meta-learning methods (Nam et al., 2023), in the few-shot learning scenario by extracting effective knowledge from the transfer sources via prompting. Finally, in Section 4.3, we validate the effectiveness of our proposed *pseudo-demonstration* and that better performance can be achieved with a more advanced LLM (*i.e.*, GPT-4; OpenAI, 2023).

**Common setup and baselines.** When using unlabeled data as the training source, we use 20% of the labeled dataset for test samples and convert the remaining 80% to unlabeled, except for a limited number of labeled samples. Following Nam et al. (2023), we one-hot encode categorical features for the baseline and then min-max scaling. To validate P2T, we consider supervised learning baselines such as CatBoost (Prokhorenkova et al., 2018), logistic regression (LR), and nearest neighbor classifier (kNN) that do not utilize unlabeled (or heterogeneous) data. We also consider VIME (Yoon et al., 2020), a self-supervised learning baseline where the model is initially pre-trained and then evaluated using labeled samples via logistic regression. Note that we intentionally exclude methods that require careful hyperparameter tuning, *e.g.*, MT (Tarvainen & Valpola, 2017), MPL (Pham et al., 2021), ICT (Verma et al., 2022), due to sensitivity to hyperparameters and overfitting issues; our problem settings characterized by limited labeled data are not suitable for hyperparameter tuning in real-world scenarios due to the lack of labeled validation sets. We also consider

| Dataset | LR | kNN | CatBoost | VIME | STUNT | LIFT-ICL | **P2T (Ours)** |
|---|---|---|---|---|---|---|---|
| | | | | # shot = 1 | | | |
| Breast | 61.23 | 61.88 | 57.64 | 57.38 | 53.04 | 66.43 | **68.93**±6.13 |
| TAE | 37.35 | 37.26 | 34.29 | 37.87 | 36.87 | 30.97 | **43.23**±7.07 |
| Hamster | 51.07 | 51.00 | 51.87 | 51.53 | 51.73 | 48.00 | **58.67**±5.58 |
| Customers | 61.34 | 63.81 | 64.12 | 62.48 | 65.14 | 70.45 | **74.32**±6.15 |
| Pollution | 63.67 | 63.67 | 63.58 | 63.33 | 63.00 | 58.33 | **65.00**±3.73 |
| Diabetes | 57.61 | 58.56 | 58.60 | 56.95 | 61.08 | 62.60 | **68.44**±5.02 |
| Car | 36.95 | 31.51 | 32.33 | 34.51 | 36.48 | 69.13 | **71.40**±1.79 |
| BTC | 51.60 | 51.54 | 53.02 | 51.13 | 52.71 | 60.40 | **62.27**±9.05 |
| Haberman | 52.81 | 52.81 | 52.82 | 51.55 | 53.82 | 60.32 | **61.29**±5.59 |
| Caesarian | 62.50 | 62.50 | 56.63 | 60.38 | 60.06 | 55.00 | **63.75**±5.23 |
| VC | 53.76 | 53.77 | 54.00 | 56.34 | 62.11 | 70.00 | **70.64**±0.89 |
| Salaries | 59.52 | 58.18 | 58.45 | 66.55 | 70.26 | 45.53 | **71.06**±1.97 |
| Average | 54.12 | 53.87 | 53.11 | 54.17 | 55.53 | 58.10 | **64.92** |
| | | | | # shot = 5 | | | |
| Breast | 61.21 | 62.33 | 57.63 | 60.89 | 61.30 | 67.86 | **72.85**±1.96 |
| TAE | 43.42 | 44.65 | 39.71 | 42.84 | 40.77 | 35.48 | **45.81**±1.44 |
| Hamster | 51.60 | 54.53 | 56.33 | 52.80 | 52.87 | 58.67 | **64.00**±7.60 |
| Customers | 60.82 | 64.92 | 81.40 | 66.07 | 66.44 | 78.41 | **83.18**±0.95 |
| Pollution | 73.33 | 72.83 | 70.58 | 75.50 | 70.92 | 65.00 | **76.67**±3.73 |
| Diabetes | 64.19 | 67.32 | 64.94 | 64.29 | 69.88 | 69.20 | **71.44**±2.26 |
| Car | 53.29 | 49.62 | 46.96 | 52.37 | 51.73 | 70.81 | **72.08**±1.03 |
| BTC | 58.03 | 55.71 | 56.43 | 55.83 | 54.11 | 67.73 | **69.33**±1.76 |
| Haberman | 53.92 | 53.40 | 55.35 | 53.45 | 54.85 | 62.26 | **64.84**±2.88 |
| Caesarian | 69.56 | 64.31 | 66.25 | 64.88 | 66.75 | 65.00 | **80.00**±2.80 |
| VC | 61.66 | 61.65 | 68.00 | 62.65 | 66.66 | 70.65 | **70.97**±1.98 |
| Salaries | 70.87 | 71.38 | 66.38 | 74.82 | **76.86** | 55.65 | 75.06±1.70 |
| Average | 60.16 | 60.22 | 60.83 | 60.53 | 61.10 | 63.89 | **70.52** |

Table 2: **Few-shot test accuracy (%) using unlabeled samples as transfer source.** # shot indicates the number of labeled samples per class. For the baselines, we report the average test accuracy over 100 different seeds. We report the average accuracy and standard deviation over 5 different seeds for LIFT-ICL (Dinh et al., 2022) and our method, due to the high cost of OpenAI API. The bold denotes the highest average score.

STUNT (Nam et al., 2023), a state-of-the-art few-shot tabular learning method. Finally, we consider LIFT (Dinh et al., 2022) in the ICL setting (LIFT-ICL) as a representative way to exploit the power of LLM. In all experiments using LLM, we use `gpt-3.5-turbo`.

### 4.1 Zero-shot prediction

One of the distinct advantages of using LLMs is that they can easily obtain the desired answer in a zero-shot manner. In fact, Hegselmann et al. (2023) have shown that LLMs can perform zero-shot prediction tasks even in tabular domains. In this section, we investigate whether the proposed P2T framework can improve the performance of zero-shot prediction by transferring knowledge from unlabeled and heterogeneous datasets, respectively. We emphasize that P2T can be seen as a *zero-shot transfer module* because it is based on the ICL.

As shown in Table 1, using a transfer source improves the zero-shot prediction performance. We first found that LLM (GPT-3.5 in our case) often fails to predict with zero-shot. For example, on the Credit-g dataset, which is a binary classification, LLM scores behind random guessing (see Table 1). However, using heterogeneous sources (*i.e.*, Credit-A) or unlabeled samples of the Credit-g dataset, P2T improves zero-shot prediction accuracy by 16.0% and 24.0%, respectively (note that we randomly select 30 samples from the transfer sources due

| Target | Source | Method | Number of samples from a source dataset ($N$) | | | | | |
|--------|--------|--------|------|------|------|------|------|------|
| | | | $N = 0$ | $N = 2$ | $N = 4$ | $N = 6$ | $N = 8$ | $N = 10$ |
| Adult | Credit-R | LR[†] | 54.00 | 69.33 | 69.33 | 66.67 | 62.00 | 57.33 |
| | | kNN[†] | 54.00 | 72.00 | 72.00 | 57.33 | 57.33 | 57.33 |
| | | CatBoost[†] | 56.00 | 54.67 | 60.00 | 61.33 | 51.33 | 49.33 |
| | | LIFT-ICL | 69.33 | 25.33 | 35.33 | 52.00 | 60.00 | 43.33 |
| | | **P2T (Ours)** | **74.67** | **75.33** | **76.00** | **77.33** | **79.33** | **80.00** |
| | Electricity | LR[†] | 54.00 | 54.67 | 50.67 | 50.00 | 37.33 | 60.00 |
| | | kNN[†] | 54.00 | 57.33 | 42.67 | 42.67 | 28.00 | 42.67 |
| | | CatBoost[†] | 56.00 | 50.00 | 50.67 | 48.67 | 45.33 | 58.00 |
| | | LIFT-ICL | 69.33 | 60.67 | 64.67 | 63.33 | 58.67 | 54.00 |
| | | **P2T (Ours)** | **74.67** | **80.00** | **76.00** | **78.67** | **80.00** | **81.33** |
| Credit-g | Credit-A | LR[†] | 52.67 | 49.33 | 48.00 | 34.00 | 42.00 | 38.67 |
| | | kNN[†] | 52.67 | **58.67** | 41.33 | 41.33 | 41.33 | 24.00 |
| | | CatBoost[†] | **55.33** | 46.67 | 41.33 | 46.67 | 40.67 | 44.00 |
| | | LIFT-ICL | 42.67 | 49.17 | 48.17 | 45.83 | 46.00 | 48.67 |
| | | **P2T (Ours)** | 55.00 | 54.50 | **58.67** | **59.33** | **59.33** | **60.67** |

Table 3: **1-shot test accuracy (%) using heterogeneous data as transfer source.** We report average test accuracy over three different seeds for all methods. Experiments for non-LM baselines (†) are implemented by extending columns for the heterogeneous data with zero-padded values. Bold indicates the highest average score.

to the high cost of the OpenAI API). We also note to provide intuition that heterogeneous datasets should come from similar domains. For example, the Credit-g dataset and the Credit-A dataset are both financial datasets consisting of an individual's credit information.

## 4.2 Few-shot prediction

For few-shot tabular prediction, we evaluate the performance when one and five labeled samples are available per class, respectively. Firstly, we found that taking few-shot labeled tabular data, serializing it into text and prompting it to LLMs (*i.e.*, LIFT-ICL in Table 2) outperforms existing state-of-the-art methods for few-shot tabular prediction (*e.g.*, STUNT; Nam et al., 2023). Thus, using LLMs as prediction models is indeed a promising direction for tabular learning, and the P2T further extends this line of work by using LLMs as transfer modules via prompting. In this section, we provide few-shot tabular prediction results using (i) unlabeled data and (ii) heterogeneous data as the transfer source, respectively.

**Transfer learning utilizing unlabeled dataset.** As demonstrated in Table 2, P2T significantly and consistently improves the few-shot prediction performance on 12 tabular datasets by utilizing unlabeled data of the same dataset as transfer source (here, we use 30 unlabeled samples that are closest to each labeled sample in terms of Euclidean distance). Note that this improvement is achieved without model updates. To provide a specific example, P2T significantly outperforms LIFT-ICL in 1-shot classification, raising the average performance from 58.10% to 64.92%. Additionally, P2T consistently achieves superior results, yielding the highest score in all 12 datasets in the 1-shot classification problem, and in 11 out of the 12 datasets in the 5-shot scenario. These results represent an improvement of approximately 6.8% and 6.6% over the best performing baselines, respectively. The significant improvement is achieved by constructing effective pseudo-demonstrations from the unlabeled dataset, therefore extracting valuable information in an in-context manner.

**Transfer learning utilizing heterogeneous dataset.** We next demonstrate the effect of introducing training samples from heterogeneous data sources. First, we found that merging distinct column sets from diverse sources in the tabular domain demands a heuristic process to create a unified feature set. Such an approach may not generalize well, and

require sophisticated designs for different data combinations. In this regard, we find our tabular serialization to be simple and effective method for combining columns from various heterogeneous sources. As tabular data is transformed into natural language, the language model can automatically understand the relations between different features from their descriptions. To investigate the effect of incorporating heterogeneous data, we consider a transfer scenario where the target data should be classified given 1-shot training samples from the same dataset and $N$ additional samples from a heterogeneous source dataset that contains disparate column sets (*e.g.*, 'work class,' 'education,' etc. for the Adult dataset and 'loan amount,' 'credit history length,' etc. for the Credit-R dataset in Table 3).

As shown in Table 3, P2T consistently benefits from heterogeneous data source. For example, 74.67%→80.00% on the Adult dataset, when 10 additional samples from the Credit-R dataset is provided. More importantly, P2T is the only method that shows steady performance improvements as the number of heterogeneous training samples $N$ increases, while the baselines are not able to properly learn from the additional samples, and their performance could even deteriorate compared to their 1-shot performances. Interestingly, cramming the extra columns from the heterogeneous dataset in LIFT-ICL only incurred noise to the accuracy. We attribute this to that our P2T framework enables the LLMs to exploit the deeper relationship between heterogeneous data, while the naïve concatenation without proper descriptions only perplexes the LLMs.

### 4.3 Ablation studies

In this section, we perform further analysis of the proposed P2T framework. (i) First, we use the column feature in the source dataset that is most relevant to the target task label as useful target for creating *pseudo-demonstrations*. Here, we ask whether the created *pseudo-demonstrations* are really similar to the real task, *i.e.*, if we create *pseudo-demonstrations* with an arbitrary column feature of the source data as a target, the created demonstrations may not be similar to the real task and hence the LLM may not convey knowledge well. (ii) Second, a variety of LLMs have emerged in recent years. This raises the question of whether P2T can achieve better performance with more advanced models.

**Effectiveness of identifying target for pseudo-demonstration construction.** In Figure 3, we validate the effectiveness of using the identified column feature as an useful target for creating *pseudo-demonstrations*. Here, we consider a transfer learning scenario utilizing an unlabeled dataset for 1-shot prediction. As shown in Figure 3, using the identified target that is highly correlated with the target task consistently outperforms using random targets (*e.g.*, using the 'Age' column as the target for the *pseudo-demonstrations* in Figure 1). Interestingly, using randomly selected column features as targets

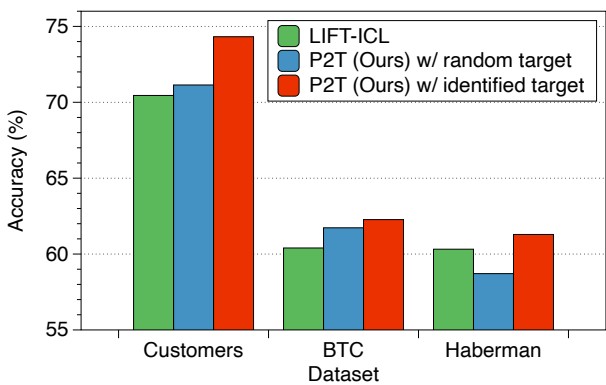

Figure 3: Ablation study that varies the column features used as targets for pseudo-demonstrations.

can perform worse than not leveraging unlabeled data (see the Haberman dataset results in Figure 3). This is because, for example, in the Haberman dataset, 'patient's year of operation' may not have any correlation to the target task 'patient's survival status.' This highlights that carefully constructing *pseudo-demonstrations* designed to be highly relevant to the target task is a key factor in enabling transfer learning via prompting.

**Qualitative analysis on selected column.** In this part, we use the Breast dataset as an example to qualitatively analyze which column feature is most relevant to breast cancer. For the Breast dataset, 'deg-malig' is selected as the most relevant, making it a useful target when creating *pseudo-demonstrations*. On the other hand, the other feature 'tumor-size' is

| Method | Customers | | BTC | | Haberman | |
|---|---|---|---|---|---|---|
| | GPT-3.5 | GPT-4 | GPT-3.5 | GPT-4 | GPT-3.5 | GPT-4 |
| LIFT-ICL | 70.45 | 88.18 | 60.40 | 61.73 | 60.32 | 67.74 |
| **P2T (Ours)** | **74.32** | **89.77** | **62.27** | **63.47** | **61.29** | **70.32** |

Table 4: **Comparison between GPT-3.5 and GPT-4.** We report 1-shot test accuracy (%) using unlabeled samples as transfer source. We report the average accuracy over 5 different seeds. The bold denotes the highest average score.

| Dataset | LR | LIFT-ICL | **P2T (Ours)** |
|---|---|---|---|
| Laptops | $7.47_{\pm 2.00}$ | $4.22_{\pm 1.00}$ | $\mathbf{3.97_{\pm 0.99}}$ |
| Choles. | $1.56_{\pm 1.58}$ | $0.37_{\pm 0.14}$ | $\mathbf{0.34_{\pm 0.04}}$ |
| House | $\mathbf{2.31_{\pm 0.46}}$ | $2.56_{\pm 0.79}$ | $2.48_{\pm 0.77}$ |

| Dataset | CatBoost | **LLM (Ours)** |
|---|---|---|
| Customers | $69.32_{\pm 4.17}$ | $\mathbf{74.32_{\pm 3.47}}$ |
| BTC | $62.00_{\pm 8.65}$ | $\mathbf{62.27_{\pm 9.05}}$ |
| Haberman | $60.97_{\pm 5.75}$ | $\mathbf{61.29_{\pm 5.59}}$ |

Table 5: **10-shot regression of P2T using unlabeled samples as transfer source.** We report the average test mean squared errors (MSEs) and standard deviation over 5 different seeds. The bold denotes the lowest average score.

Table 6: **LLM's superiority for correlation identification.** We report 1-shot test accuracy (%) using unlabeled samples as transfer source. We report the average accuracy over 5 different seeds.

also highly correlated, but LLM recommends using 'deg-malig.' In fact, both 'deg-malig' and 'tumor-size' are important factors in predicting the occurrence of breast cancer because higher 'deg-malig' corresponds to a more aggressive cancer, and 'tumor-size' indicates a more advanced stage of cancer. However, 'deg-malig' is often considered a leading factor since it is more indicative of the outcome than size alone, as a small tumor might be highly malignant and thus more dangerous than a larger but less malignant one.

**Results of using an advanced LLM.** All previous experiments use GPT-3.5 (*i.e.*, gpt-3.5-turbo). A natural extension is to ask whether better performance can be achieved by P2T using a more advanced model. To verify whether P2T performs better with advanced LLMs, we use GPT-4 (*i.e.*, gpt-4-turbo-preview) in a transfer learning scenario leveraging an unlabeled dataset. As can be seen from LIFT-ICL in Table 4, we first note that the ICL capability of GPT-4 is better. With better ICL capabilities, our P2T framework also benefits and shows better performance than using GPT-3.5. This indicates that as LLMs continue to advance, improved performance by our P2T framework is expected with future models.

**Extending P2T to regression tasks.** While our main interest is in classification tasks, one way to naturally extend P2T to regression tasks is to prompt the LLM to output the numerical values of the regression task (in classification, the LLM is guided with multiple choices). Here, we evaluate P2T's capabilities in 10-shot regression tasks on three tabular regression datasets from OpenML and Kaggle. As shown in Table 5, the results indicate that P2T is a competitive approach for few-shot tabular regression tasks. However, the improvement is relatively small compared to classification tasks. This is because LLM often faces difficulties when dealing with numerical values, and we leave it as a future work to address this.

**Using LLM for correlation identification.** To verify that LLM is better than conventional methods for identifying the most correlated feature, we perform an ablation study using CatBoost for correlation identification. As shown in Table 6, using the column feature selected by CatBoost (which differs from LLM's selection) to construct the pseudo-demonstration performs worse than using LLM. This is because (i) CatBoost often does not perform well with few-shot examples (see Table 2), and (ii) LLM, on the other hand, can understand the linguistic context of the task and thus is better able to identify correlations.

**Robustness to missing values.** In practice, tabular data often contains missing values for various reasons. For instance, biopsy results may not be collected for all patients due to the risks and complications involved in the data collection process (Yoon et al., 2018). Conventionally, missing values are managed using imputation algorithms (Yoon et al., 2018;

| Method | Customers | | BTC | | Haberman | |
|---|---|---|---|---|---|---|
| | ● → ◐ | | ● → ◐ | | ● → ◐ | |
| kNN | 63.81 | → 56.17 | 51.54 | → 50.05 | 52.81 | → 50.11 |
| LR | 61.34 | → 53.89 | 51.60 | → 51.37 | 52.81 | → 51.40 |
| LIFT-ICL | 70.45 | → 64.70 | 60.40 | → 56.00 | 60.32 | → 60.00 |
| **P2T (Ours)** | **74.32** | → **70.01** | **62.27** | → **56.93** | **61.29** | → **60.97** |

Table 7: **Robustness of P2T to missing values.** We report 1-shot test accuracy (%) using unlabeled samples as transfer source. We report the average accuracy over 5 different seeds. ● indicates that all features are used, and ◐ indicates that 50% of the features are randomly omitted. The bold denotes the highest average score.

Yi et al., 2020) in the tabular domain, which estimates missing values from other existing information. The performance of standard machine learning methodologies largely depends on imputation algorithms, as incorrectly estimated data could introduce severe noise.

In contrast, P2T naturally handles missing values by simply excluding these values from the input prompt. For example, if the 'Age' feature is missing, P2T serializes the table to prompt like "Insulin is 130, BMI 37.9." To verify the robustness of P2T to missing values, we simulate a scenario where 50% of features are randomly omitted. As shown in the Table 7, P2T outperforms the baselines using simple zero imputation (Yi et al., 2020), and the non-LM baselines fitted with all features. Furthermore, even when the non-LM baseline shows significant performance degradation (*e.g.*, on the Customers dataset), P2T shows a smaller performance drop. These results highlight P2T's robustness to missing values, which we attribute to the fact the P2T is not significantly affected by inaccurate estimates that the estimation algorithm can introduce.

**P2T without language descriptions.** Indeed, LLMs require explicit column descriptions to effectively exploit the language prior, but informative descriptions are often absent in tabular datasets (Asuncion & Newman, 2007). Consequently, researchers are forced to use the *generic indicator*, a prompt that is used to substitute (or pretend as) actual column names, for instance, 'feature' (Dinh et al., 2022). In Figure 4, we present the 1-shot test results of generic model where the language descriptions are removed. Specifically, we employ generic indicator 'feature' and 'output y' for the column names. Even in this case, leveraging unlabeled data with P2T significantly improves performance, demonstrating P2T's potential to handle all types of tabular data.

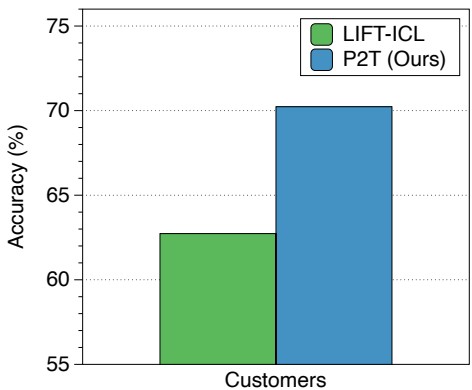

Figure 4: Transfer learning with unlabeled data without language descriptions.

## 5   Conclusion

In this paper, we introduce P2T, a novel framework for transfer tabular learning. Our approach utilizes prompting to effectively transfer knowledge from the transfer source. Through extensive experiments, we show the efficacy of P2T across various datasets. It is noteworthy that P2T does not necessitate expert-level knowledge in machine learning. This allows practitioners to adopt our framework effortlessly. We hope that our work holds promising potential in broadening the use of LLMs for tabular learning, particularly in conjunction with readily available resources such as unlabeled data.

## Acknowledgements and disclosure of funding

We would like to thank Seojin Kim, Subin Kim, Changyeon Kim, Sangkyung Kwak, Se-unghyuk Oh, and anonymous reviewers for their helpful feedback and discussions. This work was supported by Institute of Information & communications Technology Promotion (IITP) grant funded by the Korea government (MSIT) (No.RS-2019-II190075, Artificial Intelligence Graduate School Program(KAIST); No.RS-2022-II220959, Few-shot Learning of Casual Inference in Vision and Language for Decision Making) and Hyundai Motor Group.

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

# Appendix: Tabular Transfer Learning via Prompting LLMs

## A   Baseline details

In this section, we provide brief explanations of the chosen baselines. For CatBoost (Prokhorenkova et al., 2018) and logistic regression, we employ the default hyperparameters as provided by the CatBoost library and the Scikit-learn library, respectively. For VIME (Yoon et al., 2020) pre-training, we adopt the hyperparameters recommended in the original paper, utilizing the Adam optimizer with a learning rate of $1e-3$ and weight decay of $1e-4$. When implementing STUNT (Nam et al., 2023), we follow the unsupervised validation scheme proposed in the original paper for hyperparameter search and early stopping. For LIFT-ICL (Dinh et al., 2022), we provide prompt example used on the Customers dataset in Listing 1.

```
f'''Read a given information and questions. Think step by step, and then predict whether
↪  its value is class1 or class2. You must choose in [class1, class2]. Class1 indicates
↪  Horeca (Hotel, Restaurant, Cafe) channel, and class2 indicates Retail channel.

The dataset consists of 7 input variables: annual spending on fresh product, annual
↪  spending on milk products, annual spending on grocery products, annual spending on
↪  frozen products, annual spending on detergents and paper products, annual spending
↪  on delicatessen products, and customer's region. The output variable is the
↪  customer's channel.

Question: If the annual spending on fresh product is 583.0, annual spending on milk
↪  products is 685.0, annual spending on grocery products is 2216.0, annual spending
↪  on frozen products is 469.0, annual spending on detergents and paper products is
↪  954.0, annual spending on delicatessen products is 18.0, customer's region (1
↪  indicates Lisbon, 2 indicates Porto, and 3 indicates Other) is 1, then what is the
↪  customer's channel? Choose between [class1, class2]. Class1 indicates Horeca (Hotel,
↪  Restaurant, Cafe) channel, and class2 indicates Retail channel. Answer: class1

Question: If the annual spending on fresh product is 7823.0, annual spending on milk
↪  products is 6245.0, annual spending on grocery products is 6544.0, annual spending
↪  on frozen products is 4154.0, annual spending on detergents and paper products is
↪  4074.0, annual spending on delicatessen products is 964.0, customer's region (1
↪  indicates Lisbon, 2 indicates Porto, and 3 indicates Other) is 3, then what is the
↪  customer's channel? Choose between [class1, class2]. Class1 indicates Horeca (Hotel,
↪  Restaurant, Cafe) channel, and class2 indicates Retail channel. Answer: class2

Question: If the annual spending on fresh product is 11686.0, annual spending on milk
↪  products is 2154.0, annual spending on grocery products is 6824.0, annual spending
↪  on frozen products is 3527.0, annual spending on detergents and paper products is
↪  592.0, annual spending on delicatessen products is 697.0, customer's region (1
↪  indicates Lisbon, 2 indicates Porto, and 3 indicates Other) is 1, then what is the
↪  customer's channel? Choose between [class1, class2]. Class1 indicates Horeca
↪  (Hotel, Restaurant, Cafe) channel, and class2 indicates Retail channel. Answer:
'''
```

Listing 1: Prompt for LIFT-ICL (Dinh et al., 2022) used on the Customers dataset.

# B Prompt examples used in P2T

In this section, we provide examples of prompts used in P2T, specifically on the Customers dataset in scenario of transferring knowledge from the unlabeled dataset. In particular, we illustrate the prompts employed to identify the column feature with the highest correlation (see Listing 2), along with the prompts used during the final inference stage (see Listing 3). For the sake of brevity and due to constraints on paper length, the prompts we provide consist of merely two unlabeled samples along with a single labeled sample per class. We also provide prompt used for zero-shot prediction (see Table 1) on the Adult dataset, where the source dataset is Electricity, in Listing 4. Note that prompt used in Table 3 just requires additional labeled samples of the target dataset.

```
f'''Read a given information and questions. Think step by step, and then choose the
↪   most important feature to predict whether its value is class1 or class2. You must
↪   choose in [annual spending on fresh product, annual spending on milk products,
↪   annual spending on grocery products, annual spending on frozen products, annual
↪   spending on detergents and paper products, annual spending on delicatessen
↪   products, and customer's region].
The dataset consists of 7 input variables: annual spending on fresh product, annual
↪   spending on milk products, annual spending on grocery products, annual spending on
↪   frozen products, annual spending on detergents and paper products, annual spending
↪   on delicatessen products, and customer's region. The output variable is: Class1
↪   indicates Horeca (Hotel, Restaurant, Cafe) channel, and class2 indicates Retail
↪   channel.

Question: If the annual spending on fresh product is 7823.0, annual spending on milk
↪   products is 6245.0, annual spending on grocery products is 6544.0, annual spending
↪   on frozen products is 4154.0, annual spending on detergents and paper products is
↪   4074.0, annual spending on delicatessen products is 964.0, customer's region (1
↪   indicates Lisbon, 2 indicates Porto, and 3 indicates Other) is 3, then what is the
↪   customer's channel? Choose between [class1, class2]. Class1 indicates Horeca (Hotel,
↪   Restaurant, Cafe) channel, and class2 indicates Retail channel. Answer: class2

Question: If theannual spending on fresh product is 583.0, annual spending on milk
↪   products is 685.0, annual spending on grocery products is 2216.0, annual spending
↪   on frozen products is 469.0, annual spending on detergents and paper products is
↪   954.0, annual spending on delicatessen products is 18.0, customer's region (1
↪   indicates Lisbon, 2 indicates Porto, and 3 indicates Other) is 1, then what is the
↪   customer's channel? Choose between [class1, class2]. Class1 indicates Horeca (Hotel,
↪   Restaurant, Cafe) channel, and class2 indicates Retail channel. Answer: class1

Choose the most important feature to predict predict its value is class1 or class2.
↪   Answer:
'''
```

Listing 2: Prompt for correlation identification used on the Customers dataset.

```
f'''Read a given information and questions. Think step by step, and then predict whether
↪  its value is class1 or class2. You must choose in [class1, class2]. Class1 indicates
↪  Horeca (Hotel, Restaurant, Cafe) channel, and class2 indicates Retail channel.
The dataset consists of 7 input variables: annual spending on fresh product, annual
↪  spending on milk products, annual spending on grocery products, annual spending on
↪  frozen products, annual spending on detergents and paper products, annual spending
↪  on delicatessen products, and customer's region. The output variable is the
↪  customer's channel.

Question: If the annual spending on fresh product is 8190.0, annual spending on milk
↪  products is 6343.0, annual spending on frozen products is 1285.0, annual spending
↪  on detergents and paper products is 1901.0, annual spending on delicatessen
↪  products is 1780.0, customer's region (1 indicates Lisbon, 2 indicates Porto, and 3
↪  indicates Other) is 3, then what is the annual spending on grocery products. Answer:
↪  9794.0

Question: If the annual spending on fresh product is 3191.0, annual spending on milk
↪  products is 1993.0, annual spending on frozen products is 1730.0, annual spending
↪  on detergents and paper products is 234.0, annual spending on delicatessen products
↪  is 710.0, customer's region (1 indicates Lisbon, 2 indicates Porto, and 3 indicates
↪  Other) is 1, then what is the annual spending on grocery products. Answer: 1799.0

Question: If the annual spending on fresh product is 7823.0, annual spending on milk
↪  products is 6245.0, annual spending on grocery products is 6544.0, annual spending
↪  on frozen products is 4154.0, annual spending on detergents and paper products is
↪  4074.0, annual spending on delicatessen products is 964.0, customer's region (1
↪  indicates Lisbon, 2 indicates Porto, and 3 indicates Other) is 3, then what is the
↪  customer's channel? Choose between [class1, class2]. Class1 indicates Horeca (Hotel,
↪  Restaurant, Cafe) channel, and class2 indicates Retail channel. Answer: class2

Question: If the annual spending on fresh product is 583.0, annual spending on milk
↪  products is 685.0, annual spending on grocery products is 2216.0, annual spending
↪  on frozen products is 469.0, annual spending on detergents and paper products is
↪  954.0, annual spending on delicatessen products is 18.0, customer's region (1
↪  indicates Lisbon, 2 indicates Porto, and 3 indicates Other) is 1, then what is the
↪  customer's channel? Choose between [class1, class2]. Class1 indicates Horeca (Hotel,
↪  Restaurant, Cafe) channel, and class2 indicates Retail channel. Answer: class1

Question: If the annual spending on fresh product is 11686.0, annual spending on milk
↪  products is 2154.0, annual spending on grocery products is 6824.0, annual spending
↪  on frozen products is 3527.0, annual spending on detergents and paper products is
↪  592.0, annual spending on delicatessen products is 697.0, customer's region (1
↪  indicates Lisbon, 2 indicates Porto, and 3 indicates Other) is 1, then what is the
↪  customer's channel? Choose between [class1, class2]. Class1 indicates Horeca
↪  (Hotel, Restaurant, Cafe) channel, and class2 indicates Retail channel. Answer:
'''
```

Listing 3: Prompt example of P2T on the Customers dataset.

```
f'''Read a given information and questions. Think step by step, and then predict
↪   whether its value is class1 or class2. You must choose in [class1, class2]. Class1
↪   indicates 'less than 50k', and class2 indicates 'more than 50k'.

The dataset consists of 14 input variables: age, workclass, fnlwgt, education,
↪   education-num, marital-status, occupation, relationship, race, sex, capital-gain,
↪   capital-loss, hours-per-week, native-country. The output variable is the person's
↪   annual income.

Question:When num_rooms is 1.0, num_people is 6.0, housearea is 895.91, is_ac is 0.0,
↪   is_tv is 0.0, is_flat is 0.0, num_children is 1.0, is_urban is 0.0, amount_paid is
↪   222.22476310000002, then what is the ave_monthly_income? Answer:18822.41

Question:When num_rooms is 3.0, num_people is 2.0, housearea is 676.98, is_ac is 0.0,
↪   is_tv is 1.0, is_flat is 0.0, num_children is 2.0, is_urban is 1.0, amount_paid is
↪   711.565416, then what is the ave_monthly_income? Answer:22900.94

Question:When age is 53, workclass is Self-emp-not-inc, fnlwgt is 169112.0, education
↪   is Bachelors, education-num is 13, marital-status is Married-civ-spouse, occupation
↪   is Exec-managerial, relationship is Husband, race is White, sex is Male,
↪   capital-gain is 0.0, capital-loss is 0.0, hours-per-week is 40, native-country is
↪   Hungary, then what is the person's annual income? Class1 indicates 'less than 50k',
↪   and class2 indicates 'more than 50k'. Choices: [class1, class2].? Answer:
'''
```

Listing 4: Prompt used for zero-shot prediction on the Adult dataset by transferring knowledge from the Electricity dataset.

## C   Future work and limitations

Despite its superiority in transfer tabular learning, our method is constrained by the prompt size limit of LLMs. For example, tabular data with a large number of columns may not fully benefit from P2T, as the result of serialized tabular data can be excessively lengthy. Nonetheless, we hope that future advances in LLMs, GPT-4 (OpenAI, 2023) for instance, will mitigate this issue by accommodating larger prompt sizes. Another potential strategy could involve utilizing LLMs to subsample critical features, thereby enabling more efficient integration of serialized tables into LLMs. Building on this, we hope that our work will inspire researchers to further investigate the relatively under-explored problems associated with tabular learning via LLMs.

## D   Broader impacts

Tabular data often include privacy-sensitive or confidential features, such as social security numbers. As such, it is crucial to handle this data with care. However, P2T is also effective for managing anonymized features. For instance, our experiments indicate that even when categorical features are replaced with random alphabetical symbols, and generic indicators are used instead of actual column names, P2T still shows competitive performance (see Figure 4). Therefore, despite potential privacy concerns related to tabular classification, P2T shows promise for widespread use alongside privacy-preserving techniques.

# E  Dataset details

In this section, we provide detailed descriptions of the considered datasets chosen from the OpenML repository (Vanschoren et al., 2014) and Kaggle. We select 17 tabular datasets which have been previously used in the in-context learning experiments by Dinh et al. (2022); Manikandan et al. (2023). We provide detailed dataset description in Table 8.

| Dataset | Source | # Col. | # Num. | # Cat. | # Classes |
|---------|--------|--------|--------|--------|-----------|
| Heart-c | OpenML (49) | 13 | 6 | 7 | 2 |
| Breast | OpenML (13) | 9 | 0 | 9 | 2 |
| TAE | OpenML (48) | 5 | 1 | 4 | 3 |
| Hamster | OpenML (893) | 5 | 5 | 0 | 2 |
| Customers | OpenML (1511) | 7 | 6 | 1 | 2 |
| Pollution | OpenML (882) | 15 | 15 | 0 | 2 |
| Diabetes | OpenML (37) | 8 | 8 | 0 | 2 |
| Car | OpenML (40975) | 6 | 0 | 6 | 4 |
| BTC | OpenML (1464) | 4 | 0 | 4 | 2 |
| Haberman | OpenML (43) | 3 | 3 | 0 | 2 |
| Caesarian | OpenML (42901) | 5 | 4 | 1 | 2 |
| VC | OpenML (1524) | 6 | 0 | 6 | 2 |
| Adult | OpenML (1590) | 14 | 6 | 8 | 2 |
| Credit-g | OpenML (31) | 20 | 8 | 12 | 2 |
| Credit-R | OpenML (43454) | 11 | 8 | 3 | 2 |
| Electricity | OpenML (43588) | 8 | 8 | 0 | 2 |
| Credit-A | OpenML (29) | 15 | 6 | 9 | 2 |
| Salaries | Kaggle | 9 | 1 | 8 | 2 |

Table 8: **Dataset description.** We select 17 tabular datasets from the OpenML repository (Vanschoren et al., 2014) for evaluation. We denote OpenML dataset id in parentheses. One dataset is from Kaggle.

