# OpenReview forum: "Tabular Transfer Learning via Prompting LLMs"
_colmweb.org/COLM/2024/Conference — COLM_

### Official Review · Reviewer_VukR · 2024-05-08

**Rating:** 6
**Confidence:** 3
**Ethics Flag:** 1

**Summary:**

This paper proposes a novel framework, Prompt to Transfer (P2T), which utilizes large language models (LLMs) for tabular transfer learning. The methodology leverages in-context learning (ICL) by creating pseudo-demonstrations from unlabeled or heterogeneous source datasets to predict target tasks effectively. The paper is well-structured, offering a clear introduction to the problem and a detailed explanation of the approach. The experimental results show that P2T significantly outperforms existing methods across various datasets, to support the significance of the work.

**Reasons To Accept:**

- The P2T framework introduces a novel use of pseudo-demonstrations for tabular transfer learning, which is a significant contribution to the field of machine learning with LLMs.

- The experiments conducted show that P2T consistently outperforms traditional and contemporary methods in zero-shot and few-shot learning scenarios. This is clearly demonstrated with extensive experimental validation.

- The research addresses a real-world problem of data scarcity in tabular datasets, making the findings relevant for various applications, including finance and healthcare.

**Reasons To Reject:**

- Some aspects of the methodology, particularly how the pseudo-demonstrations are generated and utilized, could be described more transparently to enhance the reproducibility of the results.

- There is insufficient discussion on how the selection of source and target datasets might affect the generalizability of the P2T framework, which could be a concern for adopting this method in broader applications.

---

> ### Author Rebuttal · Authors · 2024-05-30
>
> Thanks for the review. We respond to each of your comments one by one regarding what follows.
>
> ----
> **Clarification on how the pseudo-demonstrations are generated and utilized.**
>
> We would like to clarify how P2T constructs a pseudo-demonstration. First, P2T identifies the column feature that is most correlated to the task. Then, the pseudo-demonstration is constructed with language descriptions of input-output pairs, where the output is the column feature selected in the previous step, and the input is the remaining column features. Finally, the pseudo-demonstration is concatenated into a single prompt with a few-shot labeled demonstration, which is fed to the LLM.
>
> In the meantime, some example prompts are provided in Appendix A, and other prompts will be released in the final version.
>
> ----
> **Discussion on the selection of source and target datasets.**
>
> As shown in Table 1, using unlabeled samples as the transfer source usually performs better than using heterogeneous datasets. This indicates that better performance can be achieved by using a dataset closer to the target dataset (unlabeled samples consist of all the column features of the target task except for the label column).
>
> To further alleviate concerns, we performed a more in-depth case study on zero-shot classification on the Heart-c dataset (see Table 1 for experimental setup). As shown in the table below, using the Diabetes dataset as the transfer source (Heart-c and Diabetes are both medical datasets) improves the zero-shot performance, while using the Electricity dataset, which is entirely different in nature from Heart-c, degrades the performance. This indicates that datasets that are too far from the target task may hinder LLM from outputting correct answers.
>
> \begin{array}{lcc}\hline
> &\text{Source}&\text{Acc.}\newline\hline
> \text{zero-shot}&-&60.0\newline
> \textbf{P2T (Ours)}&\text{Electricity}&58.3\newline
> &\text{Diabetes}&65.0\newline\hline
> \end{array}

---

> ### Author Response · Authors · 2024-06-04
> **A Gentle Reminder**
>
> Dear Reviewer VukR,
>
> Thank you very much again for your time and efforts in reviewing our paper.
>
> We kindly remind that we have only three days or so in the discussion period.
>
> We just wonder whether there is any further concern and hope to have a chance to respond before the discussion phase ends.
>
> Many thanks,
>
> Authors

---

### Official Review · Reviewer_ijuV · 2024-05-09

**Rating:** 7
**Confidence:** 4
**Ethics Flag:** 1

**Summary:**

This paper presents a technique called P2T for transfer learning in tabular data. The core idea of the technique is as follows: supposed you want to use LLMs for doing few or zero shot learning in a domain T where the target prediction is column t, and you have access to data from related but perhaps not exactly the same domain S. First you ask an LLM to identify the column in S that is most relevant to t; let's say s. Then you create few shot examples from S where the target prediction is column s. Each of these examples is essentially descriving the row in natural language. Additionally you can also add few examples from domain T. This makes your prompt. Using this prompt, you can ask the LLM to predict the value of t for any incoming example from T.

The main claim of the paper is that this technique improves prediction in domain T and outperforms techniques like LiftICL. In paticular the claims of the paper are
1. When asking the LLM for most relevant target column s, and including those examples in the prompt, improve 1 shot (i.e. 1 example per example from target domain) accuracy of the model, even if S is heterogenous from T.
2. Which column is selected for transfer from S, matters. Performance drops if you take a random column as target.
3. Including unlabeled examples from domain T also improves the performance.

**Questions To Authors:**

1. I am a bit confused by the section on including unlabeled examples. In what way are they included in the prompt? Why can't they also be included in LIFT-ICL and how does P2T even differ from LIFT-ICL in this case (since there is no source domain so to speak)? The finding that including unlabeled examples in the prompt helps has been brought up elsewhere as well (https://arxiv.org/pdf/2404.11018) and seems orthogonal to the core p2t idea itself. What am I missing here?

2. What does inclusiion of LR and kNN in table3 mean? is LR trained on source and target data combined and if so, how does transfer happen in this case (is it via just common regularization)? If so, it is perhaps better to follow the tehnique here: https://arxiv.org/pdf/0907.1815.

3. The finding that adding heterogenous source improves the performance significantly is quite surprising. I'd highly encourage the authors to release all of their prompts for this finding for independent verification (which should be trivial).

4. How does the performance change with the closeness between two datasets? What happens if the tasks are far apart?

5. If I were allowed to fine tune a model on source examples, how does the performance change?

6. How sensitive is this approach to different naming styles or conventions across the data?

**Reasons To Accept:**

1. The paper provides a simple and useful idea for transfer learning when using LLMs for tabular data prediction. The presented approach is simple, even incremental, but still useful.
2. The experiments for the most part are useful, and the results for 1-shot heterogenous transfer is particularly quite nice.

**Reasons To Reject:**

1. The openml data is particulatrly simplistic. Could the authors also try to replicate this work on TabZilla which seems more suitable for comparisons between various LLM techniques as well as LLM vs other ML techniques. https://openreview.net/pdf?id=CjVdXey4zT

2. Given the surpising nature of the outcomes, I do want to hold the bar on release of prompts and outcomes used for comparing p2t vs lift-icl e.g. so the community can help spot check the outcomes, and since it is trivial to verify the outcomes.

---

> ### Author Rebuttal · Authors · 2024-05-30
>
> Thanks for the review. For additional experiments, unless otherwise noted, we perform 1-shot classification using unlabeled data as the source (cf. Tab. 2).
>
> **P2T on TabZilla.**
> TabZilla and ours overlap (both are subsets of OpenML). Nevertheless, we considered more datasets in TabZilla to alleviate your concerns. The final draft will include more datasets.
>
> \begin{array}{lcc}\hline
> &\text{heart-h}&\text{profb}\newline\hline
> \text{CatBoost}&67.3& 50.0\newline
> \textbf{P2T}&75.5&54.5\newline\hline
> \end{array}
>
> **Releasing prompts.**
> We will release the prompts in the final draft (some are in Appendix A).
>
> **Using unlabeled data.**
> In this case, P2T constructs pseudo-demonstrations using the most correlated feature (among the input features) as a target. Here, making tasks from unlabeled data that resemble the causality of real input-output pairs is non-trivial, which is our unique contribution, making it different from LIFT-ICL.
>
> **LR, kNN in Table 3.**
> To clarify, we concatenate every column in both datasets based on the overlapping columns. Therefore, a classifier fitted on one dataset can still be meaningful on the other. Nevertheless, non-LM methods (including the suggested reference) cannot account for non-overlapping columns, while P2T can model the semantic relations between them, even if not present in the transfer source.
>
> **Ablation: Closeness of the datasets.**
> As shown in Tab. 1, using unlabeled data (close) as a source usually performs better than using heterogeneous data (far). To further alleviate concerns about tasks being far apart, we performed a zero-shot prediction on Heart-c. As shown below, using Diabetes as the source (both are medical data) improves accuracy, while using Electricity, which is entirely different, degrades the performance. This indicates that datasets that are too far from the target task may hinder LLM from outputting correct answers.
>
> \begin{array}{lcc}\hline
> &\text{Source}&\text{Acc.}\newline\hline
> \text{zero-shot}&-&60.0\newline
> \textbf{P2T}&\text{Electricity}&58.3\newline
> &\text{Diabetes}&65.0\newline\hline
> \end{array}
>
> **Fine-tune?**
> We utilize ICL to exploit increasingly large LLMs (cf. Tab. 4). Fine-tuning may result in better performance, but it may be too expensive.
>
> **Sensitivity: Naming styles.**
> Using clear language descriptions is important for performance. Performance drops when using generic indicators (cf. Fig. 4). However, even here, performance is improved compared to not using a transfer source.

---

> > ### Comment · Reviewer_ijuV · 2024-06-05
> > **somwhat helpful; rating remains unchanged**
> >
> > Your clarifications do help but some things are still not clear
> > 1. The lr/knn baselines are still not super clear and I believe they have not been optimized to the fullest extent possible for transfer learning.
> > 2. For unlabeled examples: can you provide an example of how it is used? e.g. let's say I have columns x,y,z and target column t. For unlabeled examples, column t is empty. So now you find the most correlated column to 't' amongst x,y, and z? If so, what is this column for the datasets being discussed? The presence of such a column in features does not seem like a very likely phenomenon.

---

> > > ### Author Response · Authors · 2024-06-06
> > > **Further Clarification**
> > >
> > > **Further clarification: LR and kNN.**
> > >
> > > First, we would like to clarify that we use all columns from the heterogeneous datasets for the LR and kNN. Specifically, by concatenating all columns in both datasets, the classifier trained on both the source and target dataset (as they will have the same input feature dimensions) can be applied to the few-shot target samples. All of the experiments in Table 3 assume that some sets of columns in the source and target datasets overlap, allowing a classifier that fits one dataset to make meaningful inferences on the other.
> > >
> > > Nevertheless, to further alleviate your concern, we conducted an additional experiment using the suggested technique (i.e., FA [1]). However, such an adaptation technique also exhibited similar behavior (cf. Table 3), presumably because it is not helpful in the few-shot scenario, which is our main problem setting.
> > >
> > > \begin{array}{lllcccccc}\hline
> > > \text{Target}&\text{Source}&\text{Method}&N=0&N=2&N=4&N=6&N=8&N=10\newline\hline
> > > \text{Adult}&\text{Credit-R}&\text{LR+FA [1]}&54.00&64.67&68.67&61.33&58.00&51.33\newline
> > > &&\textbf{P2T (Ours)}&\textbf{74.67}&\textbf{75.33}&\textbf{76.00}&\textbf{77.33}&\textbf{79.33}&\textbf{80.00}\newline\hline
> > > \text{Adult}&\text{Electricity}&\text{LR+FA [1]}&54.00&54.00&51.33&51.33&47.33&60.00\newline
> > > &&\textbf{P2T (Ours)}&\textbf{74.67}&\textbf{80.00}&\textbf{76.00}&\textbf{78.67}&\textbf{80.00}&\textbf{81.33}\newline\hline
> > > \text{Credit-g}&\text{Credit-A}&\text{LR+FA [1]}&52.67&50.00&48.673&34.67&46.00&47.33\newline
> > > &&\textbf{P2T (Ours)}&\textbf{55.00}&\textbf{54.50}&\textbf{58.67}&\textbf{59.33}&\textbf{59.33}&\textbf{60.67}\newline\hline
> > > \end{array}
> > >
> > > [1] Hal Daume III, Frustratingly Easy Domain Adaptation, ACL 2007.
> > >
> > > ----
> > > **Further clarification: Using unlabeled data as a transfer source.**
> > >
> > > For further clarification, we provide an example from the Diabetes dataset, where the target task is to predict whether a patient is diabetic (i.e., the target column), and the input columns are Insulin, BMI, and Age. The unlabeled samples do not have a target column, so the pseudo-demonstration consists of insulin, BMI, and Age. For example, the pseudo-demonstration will be ‘Q: If BMI is 37.9, and Age is 21, what is Insulin level? A: 64’ (see Figure 1, and note that Insulin is most highly correlated with the target column Diabetes). Our intuition here is that knowledge of predicting Insulin levels from BMI and Age could be useful for predicting patients with Diabetes from Insulin, BMI, and Age (as these are highly related tasks).

---

> > > > ### Comment · Reviewer_ijuV · 2024-06-06
> > > > **thanks!**
> > > >
> > > > please provide these clarifications in your paper. I have upgraded my rating thanks to your clarifications.

---

> > > > > ### Author Response · Authors · 2024-06-07
> > > > > **Thanks!**
> > > > >
> > > > > Dear Reviewer ijuV,
> > > > >
> > > > > We truly appreciate your acknowledgment of our rebuttal and overall positive review! We are happy to hear that our response helps to address your suggestions. In the final draft, we will add the clarifications, which we believe will further improve our manuscript.
> > > > >
> > > > > If you have any further comments and suggestions, please let us know. We are committed to improving the quality of our work, and we value your feedback.
> > > > >
> > > > > Thank you very much,
> > > > >
> > > > > Authors

---

> ### Author Response · Authors · 2024-06-04
> **A Gentle Reminder**
>
> Dear Reviewer ijuV,
>
> Thank you very much again for your time and efforts in reviewing our paper.
>
> We kindly remind that we have only three days or so in the discussion period.
>
> We just wonder whether there is any further concern and hope to have a chance to respond before the discussion phase ends.
>
> Many thanks,
>
> Authors

---

### Official Review · Reviewer_DX82 · 2024-05-10

**Rating:** 5
**Confidence:** 4
**Ethics Flag:** 1

**Summary:**

This paper proposes a framework called "Prompt to Transfer" (P2T) for performing transfer learning on tabular data using large language models (LLMs) and in-context learning. The key idea is to create "pseudo-demonstrations" from unlabeled or heterogeneous source data that are relevant to the target task, and then prompt the LLM with these demonstrations along with a few labeled examples from the target dataset. Specifically, P2T first identifies the column feature in the source data that is most correlated with the target task label. It then constructs pseudo-demonstrations where that column is treated as the prediction target using the other columns as input features. These demonstrations capture knowledge about predicting the correlated column, which can transfer to the actual target task. Experiments on various tabular datasets show P2T consistently outperforms previous methods in few-shot and zero-shot transfer learning settings.

**Questions To Authors:**

1. In the pseudo-demonstration construction, how sensitive is P2T's performance to the choice of the correlated column feature identified by the language model? What happens if the wrong column is selected?

2. The pseudo-demonstrations are constructed based on the assumption that predicting the correlated column feature is a good proxy for the original target task. However, what theoretical justification or empirical evidence do you have that this assumption holds across different types of tabular datasets? Are there scenarios where this assumption could break down?

3. The current approach relies on enumerating and scoring all columns to identify the most correlated feature. How does this scale to datasets with hundreds or thousands of columns? Have you explored more efficient strategies?

4. What about handling missing values or noisy data entries commonly presented in tabular datasets? How can the framework be extended to be more robust and handle realistic data issues?

**Reasons To Accept:**

- Tabular data is very common across domains like healthcare, finance, etc., but tabular transfer learning has received little attention compared to areas like vision and NLP. The P2T framework is specifically designed for tabular datasets. Developing effective transfer learning methods for tabular data is a significant challenge due to heterogeneity across datasets. P2T tackles this by leveraging large language models and in-context learning in an innovative way.
- Annotating tabular data is expensive. An effective transfer learning solution that can leverage unlabeled/heterogeneous sources can have substantial practical impact across many data-rich fields.

**Reasons To Reject:**

- The paper is largely empirical in nature and lacks deeper analysis of the proposed P2T framework. Without principled justifications, the work may be perceived as somewhat ad-hoc or heuristic.
- The experiments mainly focus on classification tasks (mostly binary classification) across a limited set of 17 tiny tabular datasets. Evaluating P2T on more datasets, especially larger-scale ones, as well as other tasks like regression could improve the generalizability of the results.

---

> ### Author Rebuttal · Authors · 2024-05-30
>
> Thanks for the review. For additional experiments, unless otherwise noted, we perform 1-shot classification using unlabeled data as the source (cf. Tab. 2).
>
> **Analysis, e.g., empirical evidence.**
> A common consensus for improving ICL is to use good demonstrations. Therefore, pseudo-demonstrations indeed resemble real causal relationships (cf. Fig. 3). Here, we assume that the features in the source must related to the target. Using unrelated data will break the assumption, e.g., using Electricity (entirely different from Heart-c) as the source degrades zero-shot accuracy on Heart-c from 60.0 to 58.3.
>
> **Data w/ many columns.**
> One strategy is to ask the LLM to select the important features for the task and then run P2T only with them. As shown below, with half of the features (but informative), P2T outperforms the best baseline. Therefore, P2T can be scaled efficiently to datasets with many columns.
>
> \begin{array}{lccc}\hline
> &\text{Customers}&\text{BTC}&\text{Haberman}\newline\hline
> \text{Best baseline (All col.)}&70.5&60.4&60.3\newline
> \textbf{P2T (Half col.)}&72.7&62.1&61.0\newline\hline
> \end{array}
>
> **Regression.**
> We evaluate P2T in 10-shot regression, reporting scaled MSE. As shown below, P2T is a competitive approach for a few-shot regression. However, the improvement is relatively small compared to classification. We believe this is because LLM often suffers from dealing with numerics, leaving it as a future work to address this.
>
> \begin{array}{lccc}\hline
> &\text{Laptops}&\text{House}&\text{Cholesterol}\newline\hline
> \text{kNN}&9.3&4.2&3.1\newline
> \text{LR}&7.5&15.6&2.3\newline
> \text{LIFT}&4.2&3.7&2.6\newline
> \textbf{P2T}&4.0&3.4&2.5\newline\hline
> \end{array}
>
> **Choice of the correlated feature.**
> We’ve already shown that P2T outperforms other possible choices in Fig. 3. Specifically, using random features can lead to worse performance because the wrong columns may be selected.
>
> **Handling missing values.**
> We verify P2T’s robustness to missing values, where 50% of the features are randomly omitted. Specifically, P2T naturally handles missing values by excluding them from the prompt. As shown below, P2T outperforms the baselines (using zero imputation) and shows a small performance drop when compared to LR’s 7% drop on Customers (cf. Tab. 2).
>
> \begin{array}{lccc}\hline
> &\text{Customers}&\text{BTC}&\text{Haberman}\newline\hline
> \text{LR}&53.9&51.4&51.4\newline
> \text{LIFT}&64.7&56.0&60.0\newline
> \textbf{P2T}&70.0&56.9&61.0\newline\hline
> \end{array}

---

> ### Author Response · Authors · 2024-06-04
> **A Gentle Reminder**
>
> Dear Reviewer DX82,
>
> Thank you very much again for your time and efforts in reviewing our paper.
>
> We kindly remind that we have only three days or so in the discussion period.
>
> We just wonder whether there is any further concern and hope to have a chance to respond before the discussion phase ends.
>
> Many thanks,
>
> Authors

---

### Official Review · Reviewer_kaWX · 2024-05-12

**Rating:** 7
**Confidence:** 3
**Ethics Flag:** 1

**Summary:**

The paper presents a novel framework, Prompt to Transfer (P2T), which leverages large language models (LLMs) for transfer learning in the domain of tabular data. The authors address a significant problem in machine learning applications where labeled data is scarce, and they propose a solution that creates pseudo-demonstrations from unlabeled or heterogeneous source data to enhance the learning process. The paper claims that P2T outperforms existing methods on various tabular learning benchmarks.

**Questions To Authors:**

1) How many test samples are there in each dataset?
2) The experiments in the paper only involve the case where the transfer source is a single dataset. Is it feasible to use multiple datasets?

**Reasons To Accept:**

1) The paper introduces a unique approach to transfer learning in tabular data by utilizing LLMs, which is a novel contribution to the field.
2) The authors provide extensive experimental results demonstrating the effectiveness of P2T across diverse datasets, which supports the significance of their work.
3) The method's ability to operate without the need for training makes it accessible to environments with constrained hardware resources.

**Reasons To Reject:**

1) The proposed method in the paper is too simplistic and does not fully exploit the capabilities of large language models. There is a lack of exploration into the characteristics of tabular data.
2) The method in the paper requires calling third-party APIs, which raises privacy concerns in real-world applications. It is suggested that the authors consider using open-source large language models.
3) While the effectiveness of using Pseudo-demonstrations has been verified, there should be more attempts and comparative analyses in constructing Pseudo-demonstrations.
4) The section on Correlation Identification needs to experiment with conventional algorithms to demonstrate the superiority of using large language models.

---

> ### Author Rebuttal · Authors · 2024-05-30
>
> Thanks for the review. For additional experiments, we perform 1-shot classification using unlabeled data as the source (cf. Table 2).
>
> **How does P2T exploit the capabilities of LLMs?**
> P2T mainly consists of three components to thoroughly exploit LLM’s capabilities. First, the LLM processes heterogeneous tables within a unified language space. Second, we leverage ICL to extract transferable knowledge from the source data. Finally, we carefully design our prompts by referring to some prompting skills (e.g., using “Think step by step.”).
>
> **Result with Llama 3 70B, an open-source LLM.**
> The table below shows the compatibility of P2T with open-source LLMs.
>
> \begin{array}{lccc}\hline
> &\text{Customers}&\text{BTC}&\text{Haberman}\newline\hline
> \text{Best non-LM baseline}&65.1&53.0&53.8\newline
> \textbf{P2T (Ours)}&72.1&53.6&60.0\newline\hline
> \end{array}
>
> **More attempts on pseudo-demonstrations.**
> We carefully note that we have already considered some design choices in Figure 3. Specifically, using the random column feature as the target of the pseudo-demonstrations performed worse than our design. Such a suboptimal design can make P2T perform worse than not utilizing the transfer source.
>
> **LLMs’ superiority for correlation identification.**
> We tried CatBoost for comparison. As shown in the table below, using the column feature selected by CatBoost (which differs from LLM’s selection) to construct the pseudo-demonstration performs worse. We believe that this is because (i) CatBoost often does not perform well with few-shot examples (cf. Table 2), and (ii) LLM, on the other hand, can understand the linguistic context of the task.
>
> \begin{array}{lccc}\hline
> \text{Identification}&\text{Customers}&\text{BTC}&\text{Haberman}\newline\hline
> \text{CatBoost}&69.3&62.0&61.0\newline
> \textbf{LLM (Ours)}&74.3&62.3&61.3\newline\hline
> \end{array}
>
> **Number of test samples.**
> We use 20% of the dataset as test samples (cf. Section 4). The exact values will be included in the final revision.
>
> **Using multiple datasets.**
> P2T can be extended to use multiple datasets by constructing pseudo-demonstrations from each transfer source. However, the datasets need to be closely related so that benchmarking will take time. Therefore, we will try our best to incorporate results using multiple datasets in the final revision.

---

> ### Author Response · Authors · 2024-06-04
> **A Gentle Reminder**
>
> Dear Reviewer kaWX,
>
> Thank you very much again for your time and efforts in reviewing our paper.
>
> We kindly remind that we have only three days or so in the discussion period.
>
> We just wonder whether there is any further concern and hope to have a chance to respond before the discussion phase ends.
>
> Many thanks,
>
> Authors

---

### Decision · Program_Chairs · 2024-07-10

**Decision:**

Accept

**Comment:**

This paper proposes a simple yet effective method that uses LLMs for transfer learning on tabular tasks. The main idea is to create "pseudo-demonstrations" identify from unlabeled source data and then prompt the LLM with these demonstrations + (optionally) labeled examples from the target dataset to do the final task. Experiments on a range of tables and tasks from OpenML demonstrate the effectiveness of the proposed method.

Reasons to Accept
- Transfer learning on tabular data is important due to the scarcity and cost of labeled data. Using LLMs in the proposed manner is a plausible approach.
- Reasonably well-written paper.

Reasons to Reject
- A major criticism, as pointed out by DX82 and ijuV and I also concur, is the simplicity and small size of the tabular tasks being used. Tabular ML has come a long way and there are many datasets and tasks that have a level of complexity closer to real-world application scenarios. See TURL: Table understanding through representation learning and references therein and thereof for some pointers. As it currently stands, it's not clear how well the proposed method will generalize to real-world tables and tabular tasks that could be more complex.